# Annually Urban Fractional Vegetation Cover Dynamic Mapping in Hefei, China (1999–2018)

**Yuliang Wang** [1,2] and **Mingshi Li** [1,3,*]

1 College of Forestry, Nanjing Forestry University, Nanjing 210037, China; ylw@chzu.edu.cn
2 School of Computer and Information Engineering, Chuzhou University, Chuzhou 239000, China
3 Co-Innovation Center for Sustainable Forestry in Southern China, Nanjing Forestry University, Nanjing 210037, China
* Correspondence: nfulms@njfu.edu.cn

**Abstract:** Vegetation measures are crucial for assessing changes in the ecological environment. Fractional vegetation cover (FVC) provides information on the growth status, distribution characteristics, and structural changes of vegetation. An in-depth understanding of the dynamic changes in urban FVC contributes to the sustainable development of ecological civilization in the urbanization process. However, dynamic change detection of urban FVC using multi-temporal remote sensing images is a complex process and challenge. This paper proposed an improved FVC estimation model by fusing the optimized dynamic range vegetation index (ODRVI) model. The ODRVI model improved sensitivity to the water content, roughness degree, and soil type by minimizing the influence of bare soil in areas of sparse vegetation cover. The ODRVI model enhanced the stability of FVC estimation in the near-infrared (NIR) band in areas of dense and sparse vegetation cover through introducing the vegetation canopy vertical porosity (VCVP) model. The verification results confirmed that the proposed model had better performance than typical vegetation index (VI) models for multi-temporal Landsat images. The coefficient of determination ($R^2$) between the ODRVI model and the FVC was 0.9572, which was 7.4% higher than the average $R^2$ of other typical VI models. Moreover, the annual urban FVC dynamics were mapped using the proposed improved FVC estimation model in Hefei, China (1999–2018). The total area of all grades FVC decreased by 33.08% during the past 20 years in Hefei, China. The areas of the extremely low, low, and medium grades FVC exhibited apparent inter-annual fluctuations. The maximum standard deviation of the area change of the medium grade FVC was 13.35%. For other grades of FVC, the order of standard deviation of the change ratio was extremely low FVC > low FVC > medium-high FVC > high FVC. The dynamic mapping of FVC revealed the influence intensity and direction of the urban sprawl on vegetation coverage, which contributes to the strategic development of sustainable urban management plans.

**Keywords:** dynamic change mapping; fractional vegetation cover; vegetation index; vegetation canopy vertical porosity; multi-temporal Landsat data

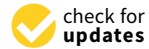



## 1. Introduction

Vegetation is considered a crucial factor in the study of global change and terrestrial ecosystems [1,2], reflecting changes, and the status of ecological environment [3]. Fractional vegetation cover (FVC) is the ratio of the vertical projection area of above-ground vegetation organs (e.g., branches, leaves, and trunks) on the ground to the total statistics areas, also called green FVC or photosynthesis FVC [4,5]. The FVC provides information on the growth status, spatial distribution, and structural changes of vegetation cover [6,7]. FVC is an essential biophysical parameter for the analysis of regional environmental change, economic development, and ecological civilization development. Particularly, change detection of FVC is widely applied to estimate changes in terrestrial ecosystems, including soil and water conservation [8,9], climate change [10–12], land use/cover change and applications [13–15], and ecosystem evaluation [16,17]. Moreover, the spatial distribution of

FVC and its spatiotemporal changes are also investigated in energy exchange calculations in different application fields, including water cycle models, vegetation photosynthesis, soil water evaporation, urban expansion, urban environment monitoring, and forest fragmentation [18,19]. Therefore, an understanding of the dynamic changes in urban FVC contributes to the sustainable development of ecological civilization in the urbanization process.

Remote sensing retrieval is an approach used to achieve FVC estimation. Remote sensing data have the characteristics of multi-resolution, flexible revisit cycles, and continuous surface imaging for wide areas, providing rich spatiotemporal information for FVC retrieval. The remote sensing method facilitates the dynamic change detection of large-scale FVC in a timely and effective manner. Numerous FVC estimation methods based on remote sensing data have been proposed and widely used [20–22]. According to the implementation method of the FVC estimation model, these methods can be divided into four categories: regression-based model, machine learning-based, mixed pixel decomposition, and index-based empirical models.

A regression model is developed between the ground-measured FVC data and the remotely sensed-based vegetation index (VI) data [23]. Linear and nonlinear regression models have been used, depending on the complexity of the relationship between FVC data and the VI [24,25]. The regression model is an empirical statistical model that is simple to implement and has high accuracy for local FVC estimation. However, its accuracy depends heavily on the ground-measured FVC, posing a considerable challenge for regional or large-scale FVC estimation [26].

Machine learning-based FVC estimation models use potential prior knowledge and information to establish the sample space by data mining. The vegetation information is extracted by sample training and iterative learning, and finally, the FVC inversion is performed. Typical algorithms include Regression Trees (RT) and Random Forest (RF) [27]. Machine learning methods have relatively high FVC detection accuracy when acquiring vegetation information. However, the FVC accuracy is influenced by remote sensing images with different spatiotemporal characteristics. The quality of remote sensing images with different spatial and temporal resolutions has differences, which affects the accuracy of vegetation and FVC. Moreover, the quality and quantity of available training samples will change with different spatiotemporal images to tuning model parameters. Additionally, machine learning algorithms have low detection efficiency from multi-temporal remote sensing images, due to high computational complexity and spatial variability [28].

A mixed pixel decomposition estimation model of FVC is based on the information contribution rate of different ground objects in each pixel. Most coarse and medium-scale resolution remote sensing images have many mixed pixels, containing spectral information on different ground objects due to the complexity and fragmentation of ground landscapes. The mixed pixel decomposition model assigns weights based on the area of different types of ground objects and their spectral responses [21,29]. Representative mixed pixel decomposition models include the dimidiate pixel (DP) model [30,31], the vegetation canopy vertical porosity (VCVP) model [32,33], the vegetation, soil, and atmospheric radiation transmission estimation (VSAR) model [34], as well as the reflectance of vegetation, lighted soil and shadow soil estimation (VISS) model. Previous studies have shown that the DP estimation model of FVC results in FVC overestimation, the VSAR and the VISS model may cause an underestimated FVC, and the VCVP model has a stable performance of FVC estimation in high- and low-density vegetation areas [35]. However, mixed pixel decomposition models depend on the reliable identification of diverse endmembers involved in the model solution, which may limit the utility of these methods in practice.

VIs can express the vegetation cover characteristics using the spectral reflectance of vegetation [36]. A VI derived from a multi-spectral image is a grayscale map, in which the brightness level indicates the degree of vegetation cover [37,38]. Typical VIs include the normalized difference vegetation index (NDVI) [39], soil adjusted vegetation index (SAVI) [40,41], modified SAVI (MSAVI) [42], optimized soil adjusted vegetation index (OSAVI) [43], and wide dynamic range vegetation index (WDRVI) [44]. These VIs facilitate

the development of vegetation extraction methods and provide an empirical model for FVC estimation. However, most VIs have disadvantages. For example, NDVI is not suitable for the estimation of FVC exceeding 60% due to its insensitivity to the change of higher 60% FVC [45]. The difference in soil reflectance may lead to a great difference in NDVI under the same FVC. The SAVI reduces interference from the soil background but reduces the correlation with FVC. Moreover, most VIs are affected by spatiotemporal changes in the remote sensing images because different reflectance of the same ground objects in different images will weaken the correlation between VIs and FVC, which lowers vegetation extraction accuracy. Thus, the accuracy of FVC estimation also fluctuates with the spatiotemporal changes of remote sensing images. Furthermore, the results of FVC estimation are closely related to urban sprawl and climatic conditions [46]. Therefore, a new VI model should be proposed to minimize the impacts of the spatiotemporal variability of remote sensing images on the accuracy improvement of VIs-based FVC estimation.

In this paper, to monitor the dynamic changes of FVC, we proposed an improved FVC estimation model by fusing an optimized dynamic range vegetation index (ODRVI) model and the vegetation canopy vertical porosity (VCVP) model. We developed a new VI to improve the accuracy of FVC estimation. The proposed VI minimizes the interference of soil and reduces the NIR reflectance in areas of high FVC, and improves the standard deviation of the gray levels in areas with high- and low-density vegetation cover. A dynamic threshold adjustment method was incorporated in the VI equation (3) to select the optimal value for multi-temporal Landsat images. The results of the ODRVI were taken as input parameters of the VCVP. Thus, the overall accuracy of FVC estimation was improved. Finally, annual FVC dynamics in Hefei, China were mapped by using the proposed models and corresponding driving factors responsible for the FVC changes were analyzed from 1999 to 2018.

## 2. Study Area and Datasets

### 2.1. Study Area

Over the past 20 years (1999–2018), Hefei, China has experienced rapid urbanization with an average growth ratio of 4%. The urbanization ratio was 76.33%, and Gross Domestic Product (GDP) grew by 7.6% in 2019. Hefei is located in the central and eastern part of China (30°57′–32°32′ N, 116°41′–117°58′ E), and has a subtropical monsoon climate, and four distinct seasons (Figure 1). The average annual temperature is about 13–17 °C, and the annual average precipitation and sunshine duration are about 800–1700 mm and 1800–2500 h, respectively. Rainfall occurs predominantly in May, June, and July, accounting for 20–38% of the annual rainfall. The months falling into a rapidly growing season for vegetation are April to October. The terrain of Hefei consists of mountains, hills, and plains, with an average elevation of about 150 m. The west mountains of Hefei are oriented from east to west, and mountains of the east are oriented from northeast to southwest, and vegetation in the south and southeast are relatively denser. The vegetation in the mountains is affected by meteorological factors, and the FVC is higher under more favorable meteorological conditions. The central part of Hefei consists of a plain, and the northern part is mainly low hills and plains where crops are grown. The vegetation cover includes large areas of farmland and some mountainous forests. Hefei was awarded the title of "National Forest City" in 2014.

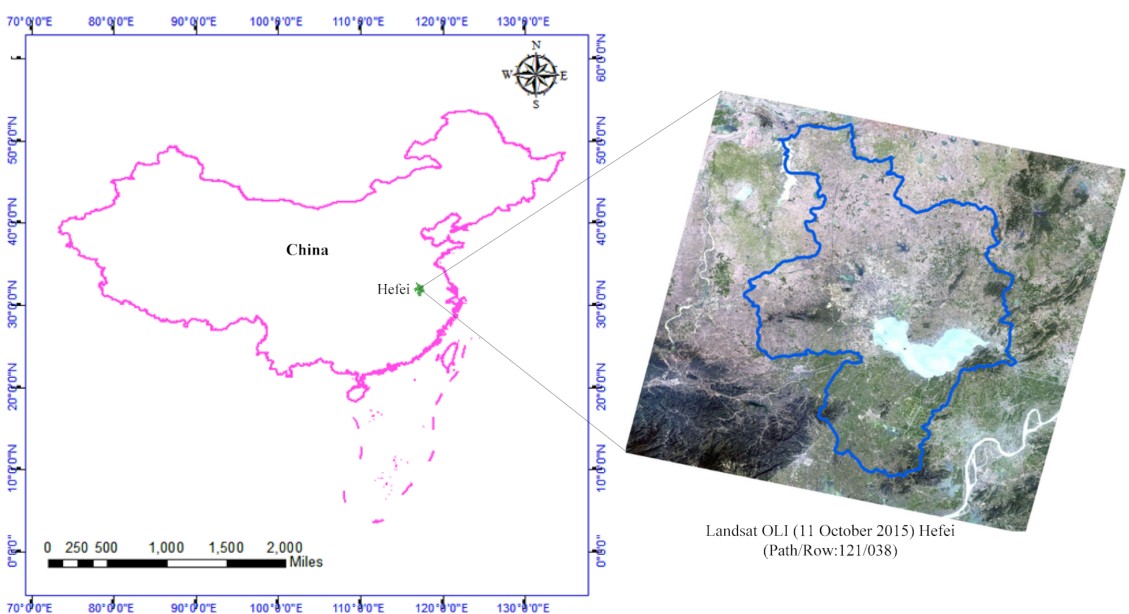

**Figure 1.** The spatial location of the study area and Landsat image (OLI: band 4 R, band 3 G, and band 2 B, WRS-2 Path/Row: 121/038). The blue polygon outlines the exact administrative boundary of the city of Hefei city.

### 2.2. Datasets

The Landsat images, with a WRS-2 Path/Row tile of 121/038, were acquired in the growing season from April to October to minimize the influence of bare soil in the harvesting and planting months. All images had a cloud cover of less than 16% (Figure 2). Cloud cover was removed during image preprocessing by using the Fmask algorithm [47]. A detailed description of the Landsat time series images from 1999 to 2018 was summarized in Figure 2. Figure 2 shows four images obtained in April, May, and August and three images obtained in July and October, respectively. The multi-temporal images were acquired by three sensors: The operational land imager (OLI) (6 Landsat 8 images), the thematic mapper (TM) (13 Landsat 5 images), and enhanced thematic mapper plus (ETM+) (1 Landsat 7 image). An automatic threshold selection algorithm was used to address the difference in the image reflectance of the three sensors [48,49]. The spatiotemporal dynamic changes of FVC in Hefei were monitored by extracting the annual FVC from the images.

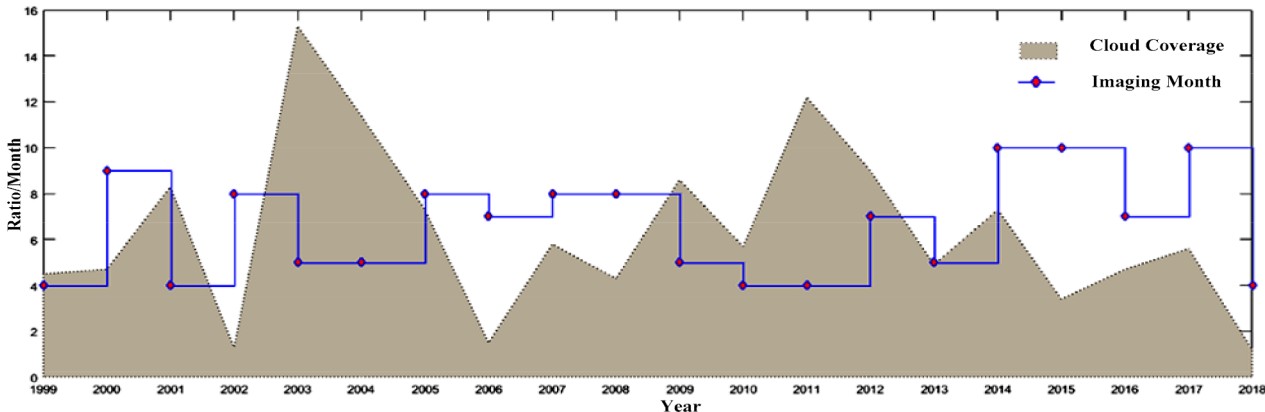

**Figure 2.** The average cloud contamination and imaging time (Month) of the selected Landsat images (TM, ETM+, and OLI).

All images were downloaded from the U.S. Geological Survey (USGS) website (https://glovis.usgs.gov/app, accessed on 18 September 2019). The data have been geometrically and atmospherically corrected by USGS to level 2 surface reflectance products. Different

numbers of images were used for each sensor due to climate effects and the imagery quality. The historical data of economic development and meteorology in Hefei (1999–2018) were collected from the Anhui Provincial Bureau of Statistics (http://data.ahtjj.gov.cn/shxfplsze/index.jhtml, accessed on 20 December 2019).

## 3. Methods

### 3.1. An Improved Vegetation Index

The spectral VIs are introduced to improve the interpretation of vegetation signals when using remote sensing data and can be used to measure vegetation status and growth while minimizing solar irradiance and soil background effects. Basically, vegetation extraction from a VI image consists of at least two steps in Landsat images, i.e., the calculation of the VI and the threshold selection to separate vegetation from other ground objects [41]. The band selection to develop a VI is the first consideration. The reflectance of an object differs in different bands. The vegetation reflectance is low in the red band and peaks in the green and NIR bands. The chlorophyll of vegetation absorbs light in the red spectrum (0.63–0.69 µm), and cellulose in vegetation foliage absorbs the NIR spectrum (0.7–1.1 µm) and the structure or compactness of cellulose highly affects the absorption rate. Therefore, the NIR and red bands were used in many VIs, such as the NDVI, SAVI, MSAVI, OSAVI, and WDRVI. The performance of these VIs was quite different in the same spatiotemporal Landsat images. The accuracy of the VIs was compared for the extraction and linear fitting results of the FVC in Hefei. The previous studies showed that the MSAVI, OSAVI, WDRVI, and SAVI had higher accuracy than other VIs, and the linear fitting between the FVC and the OSAVI and WDRVI provided the best performance [44]. Additionally, the FVC estimation accuracy for high-density vegetation and low-density vegetation was relatively stable. However, OSAVI is sensitive to soil type, roughness, and water content [43], whereas the WDRVI reduces the reflectance in the NIR band in areas of high-medium vegetation cover [44]. The bands' calculation of OSAVI is shown as follows:

$$\text{OSAVI} = (1 + L)(\rho_{NIR} - \rho_{Red}) / (\rho_{NIR} + \rho_{Red} + L) \tag{1}$$

where $\rho_{NIR}$ and $\rho_{Red}$ are the surface reflectance of the NIR band and red band, respectively. The optimum value of the parameter $L$, which reduces the effects of soil background, was set to 0.16 in the current work.

The WDRVI is calculated as follows:

$$\text{WDRVI} = (\alpha \times \rho_{NIR} - \rho_{Red}) / (\alpha \times \rho_{NIR} + \rho_{Red}) \tag{2}$$

where $\alpha$ is a weighting coefficient of the VI, and $\alpha < 1$. The WDRVI is derived from NDVI. The term $\alpha \times \rho_{NIR}$ reduces the contribution of the NIR band for FVC estimation and improves the FVC estimation stability in areas of high-medium vegetation cover.

In this paper, we proposed an improved VI that combines the strengths of OSAVI and WDRVI. The index is called the optimized dynamic range vegetation index (ODRVI). It improves the ability to distinguish between vegetation and non-vegetation, and minimizes the influence of the bare soil in areas of sparse vegetation cover. The ODRVI model optimizes the parameter settings and band operation and is defined as follows:

$$\text{ODRVI} = (1 + \theta)(\rho_{NIR} - \rho_{Red}) / (\theta \times \rho_{NIR} + \rho_{Red} + \theta) \tag{3}$$

where $\theta$ is an adjustable parameter. $\theta$ enhances the sensitivity to the soil type, roughness, and water content at the molecular level and reduces the contribution of the NIR band in areas of high-density vegetation cover in the denominator. The appropriate value of $\theta$ depends on the Landsat image type and spatiotemporal change. $\theta$ has a range of 0 to 1, like the parameter $L$ of the OSAVI [43]. The parameter $\theta$ improves the accuracy of FVC estimation in areas of high-medium and sparse vegetation covers. Verification and results

comparison was conducted to determine the parameter value, $\theta$ (empirical value) was set to 0.5 for the Hefei area in the current work.

### 3.2. VCVP–Based FVC Estimation

The VCVP model is a mixed-pixel decomposition model. The VCVP model provides FVC estimates based on the contribution rate of different cover types. Most pixels are mixed and contain different spectral responses of ground objects. The spectral structure in the pixels is calculated by the weight of the area ratio for all types of ground objects.

The VCVP model is expressed by an exponential function of the leaf area index (LAI) as follows:

$$L_0(0) = e^{-K_L LAI} \tag{4}$$

where $K_L$ is the extinction coefficient, which depends on the vegetation structure. For a given observation condition [32], most VIs can be obtained by calculating the LAI as follows:

$$VI = VI_{veg} + \left( VI_{soil} - VI_{veg} \right) e^{K_{VI} LAI} \tag{5}$$

where $VI_{veg}$ and $VI_{soil}$ are the index values of vegetation and bare soil, respectively, $K_{VI}$ depends on the solar observation angle, solar zenith angle, vegetation canopy structure, and leaf optical characteristics. The VCVP model is expressed as:

$$L_0(0) = \left( \frac{VI - VI_{veg}}{VI_{soil} - VI_{veg}} \right)^{K_L/K_{VI}}. \tag{6}$$

FVC can be expressed as 1 minus the VCVP $L_0(0)$. The FVC equation is as follows:

$$f_{veg} = 1 - L_0(0) = 1 - \left( \frac{VI - VI_{veg}}{VI_{soil} - VI_{veg}} \right)^{K_L/K_{VI}} \tag{7}$$

where $K_L/K_{VI}$ is an experimental value and a simulated value. Previous studies have shown that the FVC estimation based on the VCVP model provides stable estimates in areas of both high density and sparse vegetation cover [50]. The ODRIV was taken as parameters for the VCVP-based FVC estimation. The new equation of the FVC estimation is as follows:

$$f_{veg} = 1 - L_0(0) = 1 - \left( \frac{ODRVI - ODRVI_{veg}}{ODRVI_{soil} - ODRVI_{veg}} \right)^{K_L/K_{ODRVI}} \tag{8}$$

where $K_L/K_{ODRVI}$ is an empirical value that depends on the location of the study region and simulated values.

The FVC can be divided into different cover levels according to the vegetation cover types, spatial distribution, growth conditions, and other factors in the study area. It is divided into 5 vegetation cover grades: Extremely low FVC (ELF, 0–20%), low FVC (LF, 20–40%), medium FVC (MF, 40–60%), medium-high FVC (MHF, 60–80%), and high FVC (HF, 80–100%) [51]. The division is suitable for areas with a mixture of high density and sparse vegetation cover.

### 3.3. Automatic Threshold Selection and ODRVI Model Validation

The reflectance difference of ground objects depends on the season, spectral band, and remote sensing image types. Landsat TM, ETM+, and OLI images were used in this study. These led to an accuracy difference using a fixed threshold for multi-temporal Landsat images. A dynamic threshold selection tailored to the different images will improve the overall accuracy of vegetation extraction. Therefore, an automatic threshold selection algorithm was used to adjust the vegetation extraction from different Landsat images. An improved particle swarm optimization algorithm provides a fast search and iterative updating of the threshold to find the optimal threshold for segmentation in the global

solution space [48]. The method uses image entropy as a fitness function to obtain the optimal threshold. An index image is divided into $m$ gray levels, corresponding to a segmentation threshold $T_i$ for every gray level. The segmentation threshold dataset of the gray levels is defined as $T_v = \{T_0, T_1, T_2, \cdots, T_i, \cdots T_n\}$ $(0 \leq n \leq m, 0 \leq i \leq m)$. If $T_i$ is the optimal threshold, vegetation and non-vegetation is classified by $T_i$. The conditional equation is defined as:

$$VI_T = \begin{cases} vegetation, & Index\ value \geq T_i \\ non-vegetaion, & Index\ value < T_i \end{cases} . \qquad (9)$$

The threshold divides the index images into the foreground (vegetation) and background colors (non-vegetation). The vegetation shows light or bright gray and non-vegetation shows dark gray in index images. Dynamic threshold overcomes manual identification, fine-tuning, and testing in traditional threshold selection, and reduces accuracy interference from artificial adjustment for multi-temporal images.

We used quantitative and qualitative verification strategies to validate the ODRVI model performance and to evaluate the FVC estimation, including the relative accuracy verification, comparing with the results of convolutional neural networks (CNNs) and other typical VIs [44]. We used a CNN's deep learning algorithm to establish image supervision classification by 970 vegetation pixels, 1223 water pixels, 156 bare soil pixels, and 723 impervious surface pixels as learning samples.

## 4. Results

### 4.1. Performance of the ODRVI Model

The ODRVI model was established using Landsat surface reflectance images of Hefei, and the corresponding VI images are presented on the right line in Figure 3. Five types of ground objects were labeled and compared point by point with the original multi-spatiotemporal Landsat images, including impervious surface, bare soil, water, low vegetation cover, and high vegetation cover. Figure 3a,b shows the Landsat OLI image of Dashu Mountain Forest Park and the surrounding area of Hefei. The forest park is an area with high vegetation cover, showing bright colors with high gray values in the ODRVI image (Figure 3b). Areas of low vegetation cover are light gray in the VI image. Bare soil and impervious surfaces appear black-gray, light bright values indicate vegetation, and water tends to be dark with the lowest index value. There are many construction sites and considerable amounts of bare soils in this area, on the left side of the forest park. Vegetation, areas with sparse buildings, and high-density construction were identified in the index image (Figure 3b).

Figure 3c,d shows a Landsat ETM+ image of a wetland area of Chaohu Lake with abundant vegetation cover. Watergrass and cyanobacteria show scattered distribution from the shore to the center of Chaohu Lake. Farmland, grasslands, and forests surround the lake. Figure 3e,f shows a Landsat TM image of the rural-urban fringe of the northern part of downtown Hefei. In the ODRVI images, high-density and low-density vegetation cover have high brightness values, water has the lowest index values, and bare soil and impervious surface have intermediate index values. The ground objects are identified in the VI images, facilitating the identification of vegetation and non-vegetation. The ODRVI enhances the areas of vegetation and is suitable for all types of Landsat images. The pixel average value range of the marked five ground object types in ODRVI images was listed in Table 1.



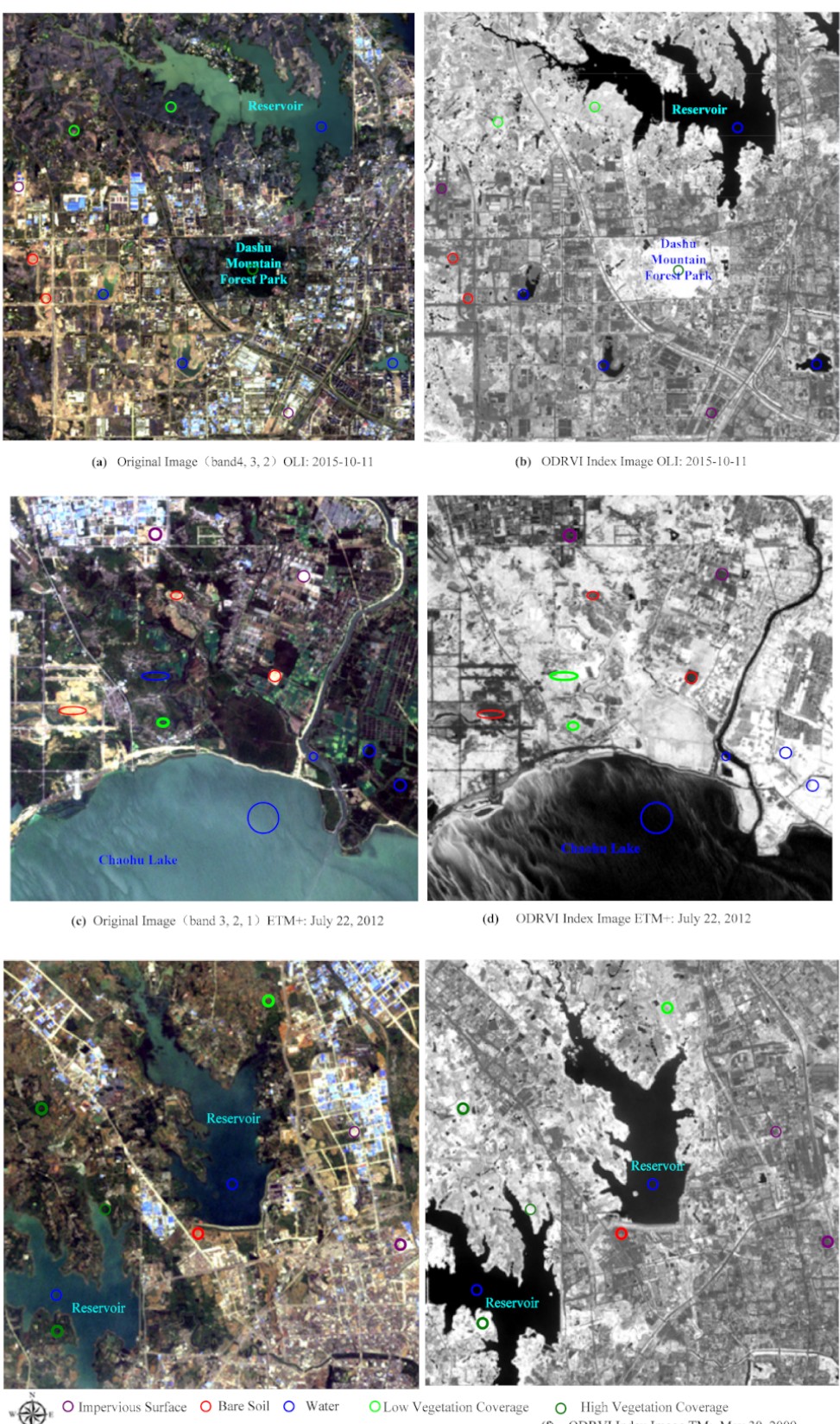

**Figure 3.** The depicting capabilities of the ODRVI model to different land cover types in different spatiotemporal Landsat images. Subfigures (**a**,**c**,**e**) are the original Landsat OLI, ETM+, and TM natural color composites, respectively. Subfigures (**b**,**d**,**f**) are the corresponding ODRVI images.

**Table 1.** The range of the marked five ground object types in ODRVI images. Impervious surface (IS), low vegetation coverage (LVC < 60%), and high vegetation coverage (HVC, >60%).

| Ground Object | IS | Bare Soil | Water | LVC | HVC |
|---|---|---|---|---|---|
| Range (±0.05) | 0.04–0.15 | 0.21–0.35 | −0.56−−0.18 | 0.98–1.32 | 1.89–2.21 |

The ODRVI image of Hefei obtained from the Landsat OLI image (25 July 2016) is shown in Figure 4. Downtown Hefei is located in the central area north of Chaohu Lake. The Dashu Mountain Forest Park with high-density vegetation cover is highlighted with a red circle in the downtown area. Forests are located west, east, and south of Hefei (marked as blue circles), and farmland is shown north of Hefei. In Figure 4b, non-vegetation is shown in low index values, water is black, and urban impervious surface areas are in dark gray. The vegetation is shown in light gray, and the higher the vegetation cover, the higher the brightness values are, such as the forests. The results show that the ODRVI facilitates distinguishing vegetation from non-vegetation.

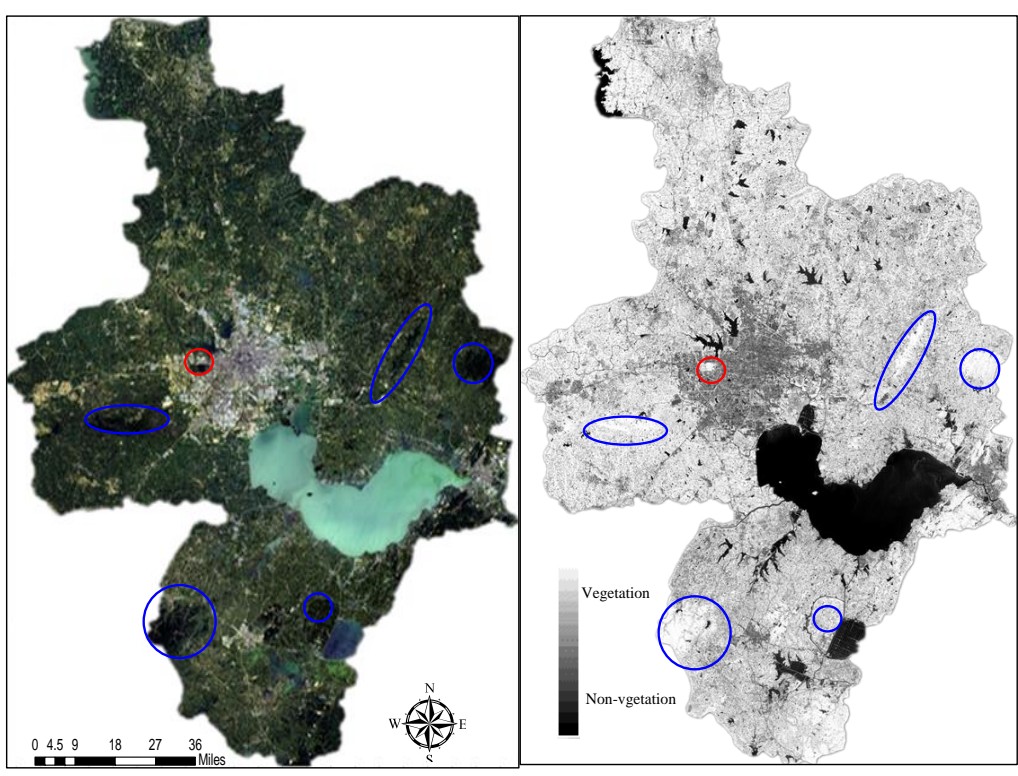

（a）Original Landsat OLI ( band 4,3,2: 2016-07-25)　　(b) ODRVI index image of Hefei（OLI: 2016-07-25）

**Figure 4.** The depicting performance of the ODRVI model derived from the Landsat OLI image in Hefei. (**a**) is the Landsat OLI image (band 4,3,2) and (**b**) is the corresponding ODRVI index image.

### 4.2. ODRVI Model Validation

The validation of the ODRVI model included an assessment of the relative accuracy of the VI obtained from the multi-spatiotemporal images and comparisons with typical VIs.

The relative accuracy of the ODRVI was verified using different spatiotemporal Landsat images, including Beijing (OLI: 10 July 2017), Hefei (OLI: 25 July 2016), and Guangzhou (OLI: 23 October 2017). The ODRVI images were achieved by the ODRVI model and the automatic threshold selection algorithm derived from spatiotemporal Landsat images. The classification accuracy obtained from the CNNs and the ODRVI images of the three regions was compared. The vegetation pixels obtained from the CNNs method were used as the reference to determine the vegetation extraction accuracy of ODRVI for the three regions. The overall classification accuracy of the CNNs for Beijing, Hefei, and Guangzhou was

96.75%, 95.68%, and 95.93%, respectively, and the classification accuracy of the vegetation class was 97.21%, 96.54%, and 97.32%, respectively. Table 2 lists the relative accuracy obtained from the CNNs and the ODRVI in the three regions. The relative accuracy of the ODRVI in Beijing, Hefei, and Guangzhou was 93.35%, 96.79%, and 95.88%, respectively.

**Table 2.** The relative accuracy of vegetation extraction of ODRVI model in different regions.

| Verification Regions | CNNs | | ODRVI | |
|---|---|---|---|---|
| | Number of Vegetation Pixels | Accuracy of Vegetation Extraction | Number of Vegetation Pixels | Relative Accuracy of Vegetation Extraction |
| Beijing | 14,981,044 | 97.21% | 13,984,601 | 93.35% |
| Hefei | 9,490,172 | 96.54% | 9,185,964 | 96.79% |
| Guangzhou | 6,100,657 | 97.32% | 5,849,127 | 95.88% |

Figure 5 shows a comparison of the results between CNNs classification and ODRVI classification for the three regions. In Figure 5d–f, the vegetation cover in Beijing, Hefei, and Guangzhou was 42.2%, 36.1%, and 37.1%, respectively. Hefei had the lowest vegetation density due to large areas of farmland north of Hefei, and Beijing had the highest vegetation density. Most of the vegetation occurred in the west and the northern mountainous areas of Beijing. The vegetation cover in the northern part of Guangzhou was higher than that in the southern part. Areas of high-density vegetation occurred in the mountainous areas, and areas of low-density vegetation were located near the city. In Figure 5g–i, a bright gray indicates vegetation cover, and non-vegetation is dark gray or black in the ODRVI image. The results showed that the ODRVI model is suitable for multi-spatiotemporal Landsat images and provides high accuracy for vegetation classification.

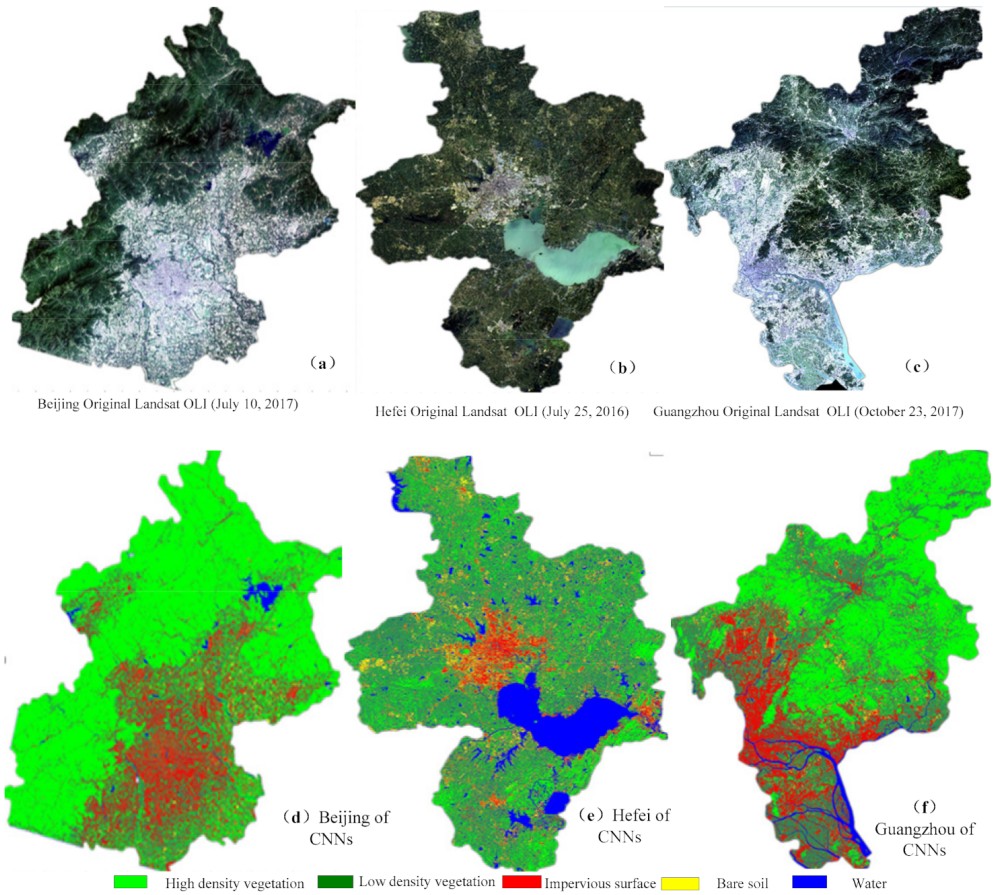

**Figure 5.** *Cont.*

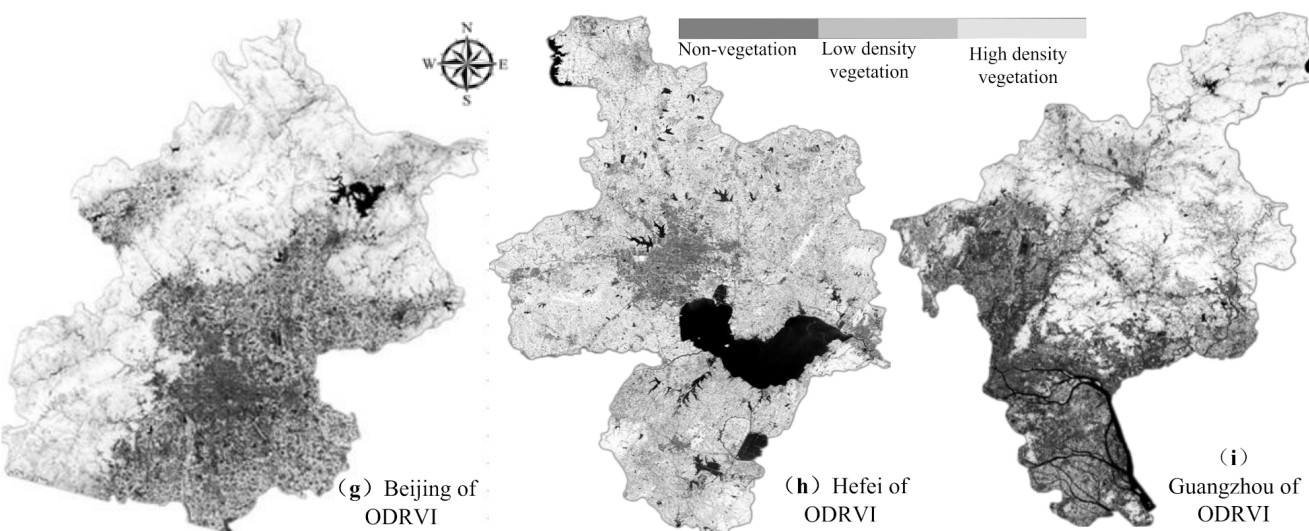

**Figure 5.** The relative accuracy verification of ODRVI over CNNs algorithm from multi-spatiotemporal Landsat images in the three regions. (**a–c**) are the original natural color images of Beijing, Hefei, and Guangzhou (bands 4, 3, and 2, stretched by 2%); (**d–f**) are the classification results of CNNs; and (**g–i**) are the ODRVI-derived results.

Moreover, the accuracy of ODRVI was further verified by the local magnified contrast of ground objects in Figure 6. There is obvious identification between vegetation and non-vegetation in the ODRVI images. The high-density vegetation and low-density vegetation show in bright gray and light gray, respectively, and the non-vegetation display dark gray or black. The circles mark the comparison areas between original images, CNNs, and ODRVI images.

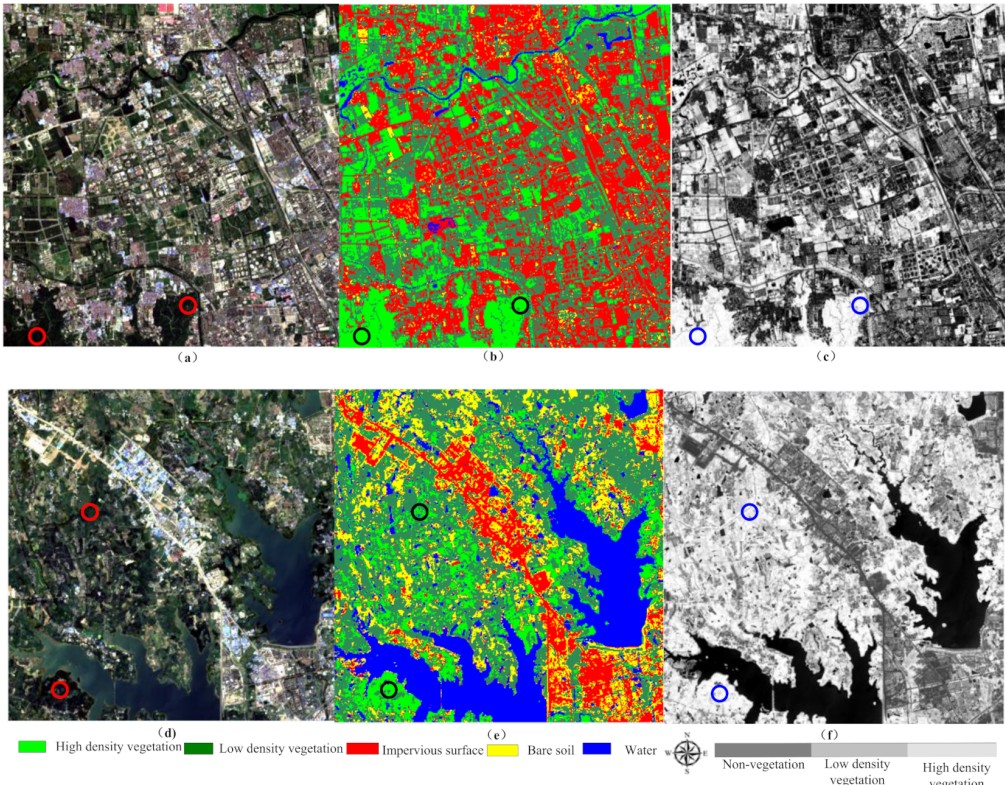

**Figure 6.** The accuracy verification of ODRVI by a local magnified contrast of ground objects in three regions. Where (**a,d**) are original images, (**b,e**) are CNNs classification, and (**c,f**) are ODRVI index results, and the ellipse shows vegetation comparison points.

In comparison to pre-existing VIs, the typical VIs being OSAVI, WDRVI, VARI, and MSAVI, ODRVI showed good performance in the extraction of vegetation from multi-temporal Landsat images. Landsat images of Beijing, Hefei, and Guangzhou acquired on 10 July 2017, 25 July 2016, and 23 October 2107, respectively, were selected. The relative accuracy based on CNNs of the typical VIs is listed in Table 3. The relative accuracy of the ODRVI was 7.24%, 7.41%, and 8.55% higher than that of the typical VIs for Beijing, Hefei, and Guangzhou, respectively. The average relative accuracy of the ODRVI was 0.91% higher than that of the OSAVI.

**Table 3.** Comparison of relative accuracy between the ODRVI and the typical vegetation indices in the three regions.

| Compared Regions | Beijing | | Hefei | | Guangzhou | |
|---|---|---|---|---|---|---|
| CNNs | Number of vegetation pixels | Accuracy | Number of vegetation pixels | Accuracy | Number of vegetation pixels | Accuracy |
| | 14,981,044 | 97.21% | 9,490,172 | 98.75% | 6,100,657 | 97.32% |
| Vegetation indices | Number of vegetation pixels | Relative Accuracy | Number of vegetation pixels | Relative Accuracy | Number of vegetation pixels | Relative Accuracy |
| OSAVI | 13,749,602 | 91.78% | 9,183,707 | 96.77% | 5,779,152 | 94.73% |
| WDRVI | 13,020,025 | 86.91% | 8,499,710 | 89.56% | 5,346,006 | 87.63% |
| MSAVI | 11,899,443 | 79.43% | 7,708,087 | 81.22% | 4,911,639 | 80.51% |
| VARI | 12,931,637 | 86.32% | 8,536,484 | 89.95% | 5,274,018 | 86.45% |
| ODRVI | 13,984.601 | 93.35% | 9,185,964 | 96.79% | 5,849,127 | 95.88% |

A comparison of gray value STD between the ODRVI and the typical VIs is shown in Table 4. The index gray value STDs of the typical VIs is lower than the ODRVI in three regions. The performance of vegetation identification from non-vegetation in the ODRVI performed best, followed by the MSAVI. Therefore, the identification performance of the ODRVI was better than other typical VIs in multi-spatiotemporal Landsat images. Compared with OSAVI, WDRVI, MSAVI, and VARI, the overall accuracy of the ODRVI model was improved by an average of 7.73% (See Table 3). The fitting degree (coefficient of determination ($R^2$)) of the ODRVI by linear regression with FVC enhanced by 7.4% on average compared to other VIs (Figure 7).

**Table 4.** Comparison of gray-value standard deviation (STD) between the ODRVI and typical VIs.

| Cities | OSAVI | WDRVI | MSAVI | VARI | ODRVI |
|---|---|---|---|---|---|
| Beijing | 0.418 | 0.447 | 0.414 | 0.255 | 0.866 |
| Hefei | 0.501 | 0.501 | 0.767 | 0.251 | 0.931 |
| Guangzhou | 0.597 | 0.559 | 0.866 | 0.226 | 0.998 |

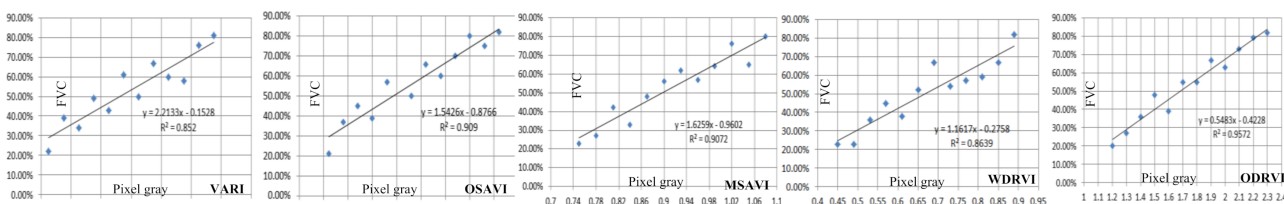

**Figure 7.** Comparison of the FVC fitting degree between ODRVI and typical VIs.

### 4.3. FVC Verification and Estimation

FVC estimation was performed using the ODRVI model and the VCVP model. The results of the ODVRI, the pure vegetation portion of a vegetation pixel $ODRVI_{veg}$, and the pure soil portion of a vegetation pixel $ODRVI_{soil}$ were input into Equation (8). The

empirical value $K_L/K_{ODRVI}$ was obtained from field measurements in Dashu and Zepeng Mountain Forest Parks in Hefei, China in 2015; the optimal value of $K_L/K_{ODRVI}$ was 0.653. We compared the results between the field measurements FVC and the proposed FVC estimation model for two sample plots in Table 5. The average accuracy of the improved FVC model was obtained at 92.97% in two sample sites. The accuracy was directly related to the vegetation growth in the current year. Experimental results showed that the average accuracy of FVC estimation would be improved in years of vegetation growing well. The average accuracy of FVC estimation in HF and MHF areas was more closed to the results of the field measurements.

**Table 5.** Accuracy comparison of the proposed FVC estimation between the two sample sites from filed measurements in Hefei (2015). Extremely low FVC (ELF), low FVC (LF), medium FVC (MF), medium-high FVC (MHF), and high FVC (HF) are listed.

| Sample Sites | Accuracy (%) | | | | |
|---|---|---|---|---|---|
| | ELF | LF | MF | MHF | HF |
| Dashu Mountain Forest Park | 93.2 | 91.5 | 92.4 | 93.1 | 94.3 |
| Zepeng Mountain Forest Park | 92.7 | 92.3 | 92.8 | 93.3 | 94.1 |

The value of the pure vegetation portion and pure soil portion of a vegetation pixel was calculated using the maximum index value $ODRVI_{max}$ and the minimum index value $ODRVI_{min}$, respectively. In this experiment, the ODRVI images obtained from the three sensors (OLI, ETM+, and TM) in four seasons were used for FVC estimation. The values of $ODRVI_{veg}$ and $ODRVI_{soil}$ in the images were calculated using the automatic threshold algorithm. The parameters of the FVC estimation equation used in the four seasons are listed in Table 6.

**Table 6.** The derived parameters for FVC estimation in four seasons of Hefei.

| Parameters | Seasons and Landsat Image Types ($K_L/K_{ODRVI}$ = 0.653) | | | |
|---|---|---|---|---|
| | Winter: OLI 15 January 2016 | Spring: TM 30 May 2009 | Summer: ETM + 22 July 2012 | Autumn: OLI 11 October 2015 |
| $ODRVI_{veg}$ | 2.13 | 2.29 | 2.57 | 2.38 |
| $ODRVI_{soil}$ | 0.839 | 0.906 | 1.275 | 0.891 |

Figure 8 shows the FVC estimation results derived from the multi-temporal Landsat images over four seasons in Hefei. The bare soil observed in the autumn was due to farmland harvest and field management, which is shown in dark gray in northern Hefei in Figure 8d. The vegetation cover is depicted in light gray and occurs predominantly in the forests in the west, east, and south of Hefei. Spring is the beginning of the growing season, and the vegetation canopy has not been closed thus, this season does not represent the real FVC (Figure 8b). Since most plants are deciduous in this area, the vegetation cover in winter also does not reflect the real FVC (Figure 8a). The outline of the downtown urban area is not well delineated in spring and summer. Summer is most suitable for FVC estimation because it is the growing peak season. The ellipses show the forests of Hefei in Figure 8, and Dashu Mountain Forest Park is shown in yellow in downtown Hefei, which is bright gray in Figure 8a,c,d. The FVC value is between 0 and 1 from ELF to HF. The HF is in bright gray and the ELF is in light gray.

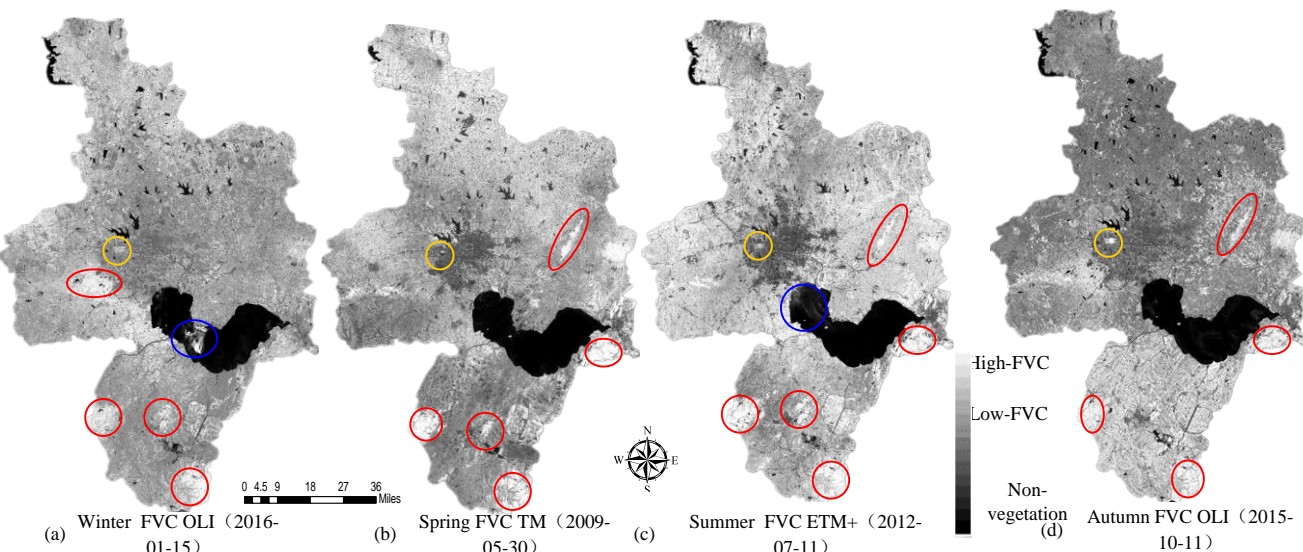

**Figure 8.** The FVC estimation results from multi-temporal Landsat images over four seasons in Hefei. (**a–d**) are FVC estimations from multi-temporal Landsat images covering four seasons.

The proposed FVC estimation was also compared with the DP-based FVC estimation in high- and low-density vegetation cover (Figure 9). Two Landsat images of Hefei with notorious differences in vegetation growth were selected: An image with lush vegetation (23 July 2006), and an image with relatively sparse vegetation cover (19 March 2010). The results showed that the medium-high and high FVC was 6.63% and 12.78% higher than the ground-truth value in the DP-based FVC estimation for the lush vegetation image, and the respective values were 2.25% and 1.57% in the VCVP-based FVC estimation. For the low vegetation cover image, there was not much difference between the DP- and VCVP-based FVC estimation, and the mean error in each FVC level was 2.45%. The VCVP-based FVC estimation exhibited a more stable performance than the DP-based FVC estimation for both high- and low-density vegetation cover, and the error was relatively low.

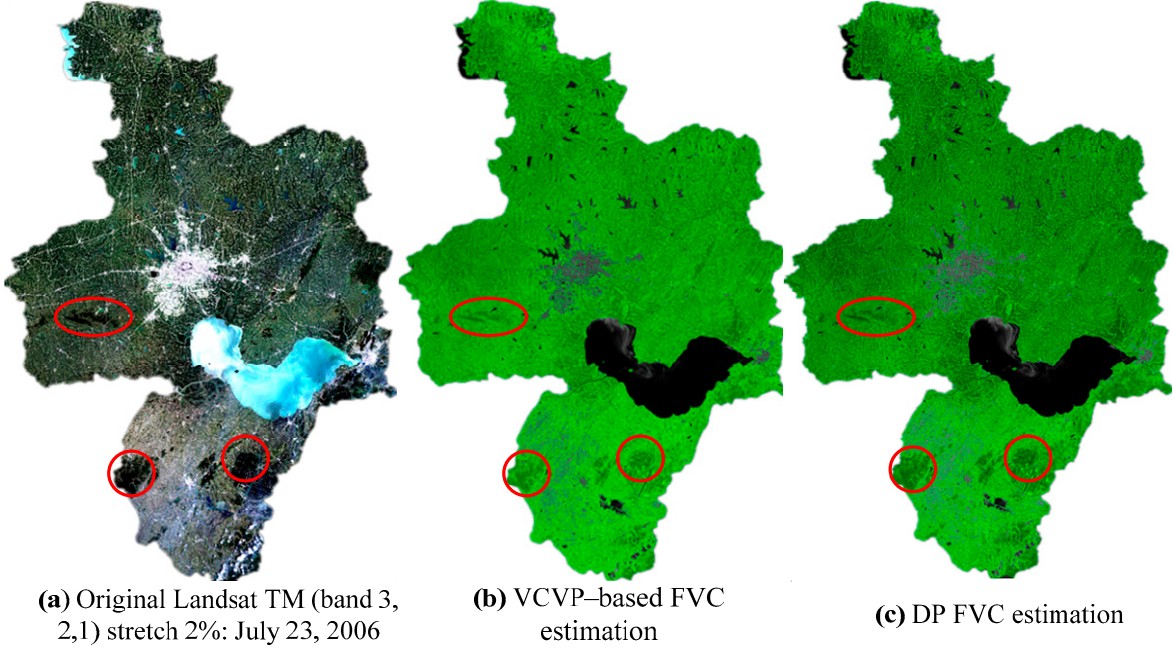

(**a**) Original Landsat TM (band 3, 2,1) stretch 2%: July 23, 2006

(**b**) VCVP–based FVC estimation

(**c**) DP FVC estimation

**Figure 9.** *Cont.*

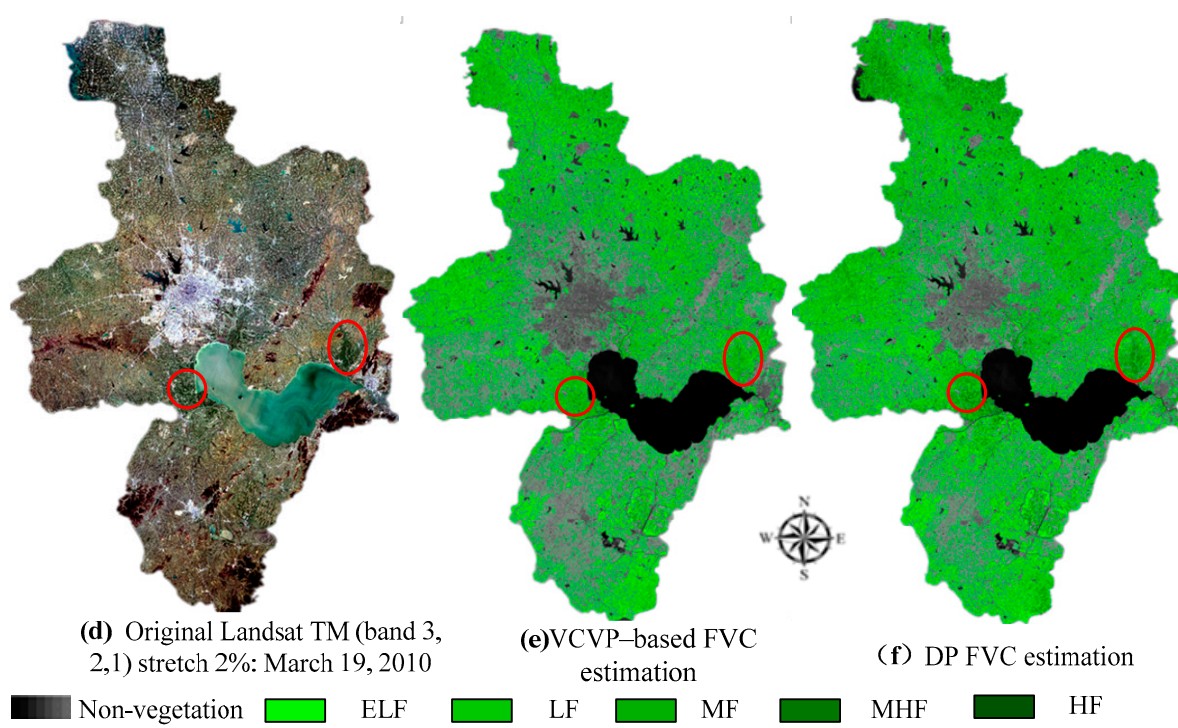

**(d)** Original Landsat TM (band 3, 2,1) stretch 2%: March 19, 2010  **(e)** VCVP–based FVC estimation  （**f**） DP FVC estimation

■ Non-vegetation ■ ELF ■ LF ■ MF ■ MHF ■ HF

**Figure 9.** Comparison of results between VCVP-based and DP FVC estimation. (**a**,**d**) are the original images, (**b**,**e**) are results of VCVP-based FVC estimation, and (**c**,**f**) are the results of DP FVC estimation. The red ellipses are the comparison points.

### 4.4. FVC Dynamic Change Mapping in Hefei (1999–2018)

The FVC dynamic change maps in Hefei derived from the VCVP-based FVC estimation from 1999 to 2018 are shown in Figure 10. The corresponding parameter values of $ODRVI_{veg}$, $ODRVI_{soil}$, and $K_L/K_{ODRVI}$ derived from this analysis are listed in Table 7. In Figure 10, areas with HF levels are in darker green and the bright green indicates an ELF. The same FVC level shows different colors over time due to the differences in the image acquisition time, surface temperature, and rainfall. In the past 20 years, the areas of MF and HF were relatively small; the HF areas accounted for about 1% of the total vegetation area, and the MHF areas comprised about 6% in Hefei. The areas of ELF, LF, and MF accounted for about 80%, among which the MF accounted for 18.7%, and the ELF comprised over 60%. However, the areas of different FVC levels varied from year to year; for example, the MF was 48.89% in 2005, the MHF was 33.23% in 2006, and the combined area of the MF and MHF FVC was 4.91% in 2014. The differences in the areas of the FVC levels were attributed to social and environmental factors, such as urban sprawl, commercial development, and changes in rainfall and temperature. We will discuss this in Section 5.

**Table 7.** Parameter selection of $ODRVI_{soil}$ and $ODRVI_{veg}$ (1999–2018) in Hefei.

| Year | $ODRVI_{soil}$ | $ODRVI_{veg}$ | Year | $ODRVI_{soil}$ | $ODRVI_{veg}$ |
|------|------|------|------|------|------|
| 1999 | 0.465 | 1.67 | 2009 | 0.906 | 2.35 |
| 2000 | 0.681 | 2.36 | 2010 | 0.836 | 2.08 |
| 2001 | 0.411 | 1.88 | 2011 | 0.811 | 2.31 |
| 2002 | 1.297 | 2.63 | 2012 | 1.275 | 2.69 |
| 2003 | 0.763 | 2.29 | 2013 | 1.153 | 2.55 |
| 2004 | 0.823 | 2.41 | 2014 | 1.052 | 2.66 |
| 2005 | 1.057 | 2.56 | 2015 | 0.891 | 2.43 |
| 2006 | 1.103 | 2.59 | 2016 | 1.117 | 2.75 |
| 2007 | 0.947 | 2.51 | 2017 | 0.897 | 2.29 |
| 2008 | 0.864 | 2.28 | 2018 | 0.963 | 2.44 |

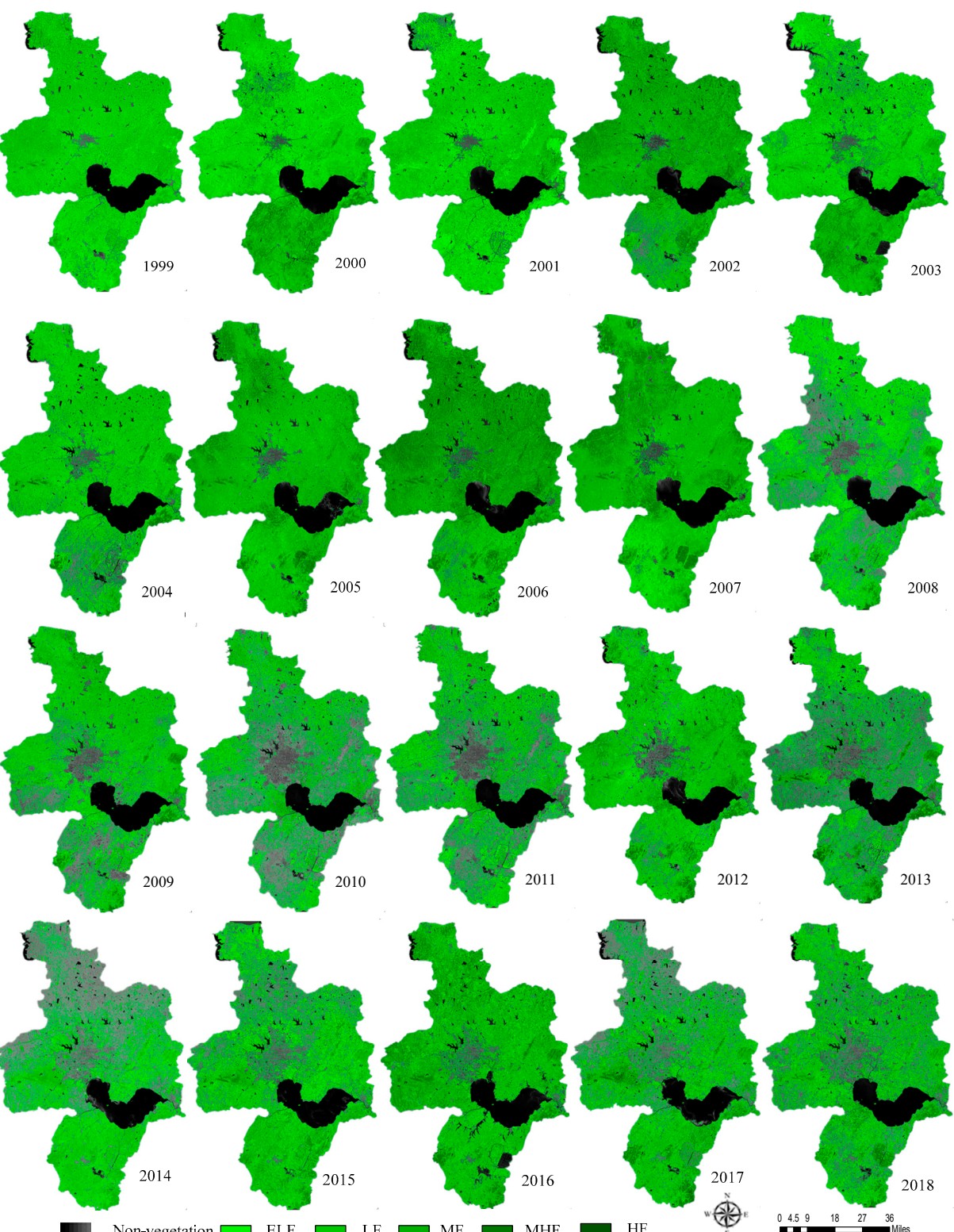

**Figure 10.** Annual FVC maps in Hefei, China, during the period 1999 to 2018. ELF, LF, MF, MHF, and HF are extremely low FVC, low FVC, medium FVC, medium-high FVC, and high FVC, respectively.

Figure 11 shows the annual area changes of each FVC level in Hefei from 1999 to 2018. The total FVC area decreased by 3265.52 km$^2$ over the past 20 years, representing a decline of 33.08%. In 1999, the total area of all FVC levels was 9872.09 km$^2$, but in 2004, the total area decreased to 7926.51 km$^2$, a decrease rate of 19.71%. From 2004 to 2010, the

total area of all FVC levels continuously dropped to 4594.06 km$^2$, a decrease rate of 42.04%. However, from 2010 to 2018, the total FVC area showed an increasing trend. In 2018, the total area of all FVC levels had increased by 2012.51 km$^2$ compared with 2010, representing an increased rate of 30.46%. The total area of all FVC levels in 2005 was 10,051.28 km$^2$. The MF accounted for 48.89% of the total area. In 2002, 2007, and 2015, the total area of all FVC levels exceeded 10,000 km$^2$, and the area of MF and MHF increased, indicating a growth of vegetation coverage in these years. In the past 20 years, the area of MF had greater fluctuation change than other FVC grades. For the other grades FVC, the order of the standard deviation of the change ratio was ELF > LF > MHF > HF.

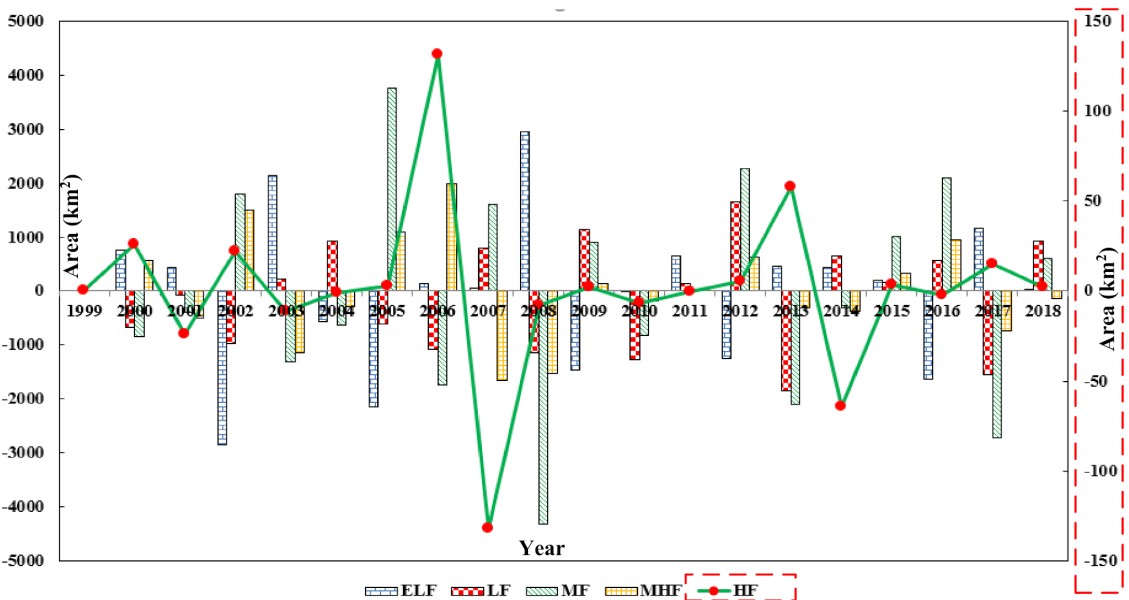

**Figure 11.** Annual area changes of each FVC grade in Hefei (1999–2018). The horizontal axis is the year, and the vertical axis is the area of change for each FVC grade. A value less than 0 represents an area decrease, and greater than 0 represents an area increase.

### 4.5. Results Analysis of FVC Changes in Hefei (1999–2018)

The total area of all FVC levels decreased from 44.74% in 1999 to 29.94% in 2018. However, the area proportion of all FVC levels fluctuated over time. The highest area share of all FVC levels was 62.34%, and the lowest was 20.82%. The FVC change was divided into four stages in the past 20 years.

(1) Continuously rapid decay (1999–2004). The total area of all FVC levels showed a rapid decline, with an average decline rate of 1.76% and a total area decrease of 1380.46 km$^2$. The area of ELF, LF, and MF decreased by 1.06%, 5.63%, and 14.29%, respectively. The area of MHF and HF in the mountains of Hefei increased by 1.15% and 11.83 km$^2$, respectively.

(2) Rapid decline with fluctuations (2005–2008). The total area of all FVC levels decreased by 3883.28 km$^2$ from 2005 to 2008, with an average decrease rate of 7.97%. From 2005 to 2008, the area of MF decreased by 45.03%, and those of LF, MHF, and HF decreased by 11.99%, 14.14%, and 0.19%, respectively.

(3) Fluctuated attenuation (2009–2013). The average vegetation cover area decreased by 382.9 km$^2$ per year, the total area of all FVC levels decreased by 5.45%, and the areas of ELF, LF, and MF decreased by 13.73%, 6.53%, and 1.61%, respectively. The area of HF increased by 0.54%. The changes in the areas of LF and MF occurred in areas surrounding the southwest, southeast, and south of downtown Hefei, and changes in the area of HF were observed in the eastern and southeastern areas.

(4)   Fluctuated increase (2014–2018). The overall area of all FVC levels showed an increasing trend (policy intervention factor), and the area increased by 1294.59 km$^2$ in Hefei. The areas of MF, LF, MHF, and HF increased by 10.02%, 1.12%, 4.22%, and 0.19%, respectively. A rapid increase in the area of 1972 km$^2$ occurred from 2015 to 2016. The areas of LF, MF, and MHF increased by 5.76%, 21.21%, and 3.39% in the north and center of downtown Hefei, respectively.

In the past 20 years, the fluctuations in the area of all FVC levels decreased, and the vegetation cover increased in some years (2005, 2009, 2012, 2016, and 2017). The changes in FVC were closely related to the current year's temperature, rainfall, sunshine duration, and other factors. The related discussions of driving factors will be shown in Section 5.

## 5. Discussion

### 5.1. Performance of the ODRVI Model

The ODRVI model comprehensively used a red band, NIR band, and a dynamic adjustment parameter $\theta$ to construct a VI model, meeting the requirements of vegetation extraction accuracy from multi-temporal Landsat images. In comparison with the OSAVI model, the ODRVI model reduced the contribution of the NIR band by parameter adjustment, which improved the FVC estimation stability in high-medium vegetation cover areas [43]. Compared with the WDRVI model, the ODRVI model enhanced sensitivity to the soil type, roughness, and water content at the molecular level, which reduced the influence from soil background [44]. Moreover, the ODRVI model applied an automatic threshold adjustment method based on the particle swarm optimization algorithm (PSO). The automatic threshold algorithm reduced the accuracy difference of vegetation extraction from multi-temporal Landsat images. The algorithm had some advantages and could be summarized as follows: (1) The PSO-based method avoided inaccuracies of coding, and achieved a higher accuracy with fewer parameters; (2) the PSO-based method used a single information-sharing mechanism, and obtained a better search speed; and (3) the image entropy method was used for the fitness function of the PSO-based algorithm, which ensured an optimal threshold for multi-temporal Landsat images [52].

The gray value identification of the ground object is a critical point of the performance of VI models. The higher the gray value standard deviation (STD) between vegetation and non-vegetation, the more accurate the identification of them [44]. The ODRVI enhanced the gray value STD via parameter configuration (see Equation (8)). The performance of the ODRVI model was verified with better identification of gray value cover in three regions (Beijing, Hefei, and Guangzhou) and different temporal Landsat images in Section 4.2 (see Table 2 and Figure 5). The overall accuracy of the ODRVI model was improved by an average of 7.73% (see Table 3), the gray value STD increased by an average of 0.45 (see Table 4), and the coefficient of determination ($R^2$) of the ODRVI by linear regression with FVC increased by 7.4% on average compared to other VIs. Therefore, the performance of the ODRVI was better than other typical VIs, meeting the needs of vegetation mapping from multi-spatiotemporal Landsat images.

However, the verification of the ODRVI model only used the surface reflectance values of ground objects in Landsat images, and no other spectral image data were tested. The same ground object imaged by different sensor types will have different reflectance values due to different spectral response functions used, which brings some challenges for accurate vegetation extraction. To overcome these challenges, further development of new approaches or improving the VI model is required. In the current study, for different optical remote sensing data, high accuracy of vegetation extraction was achieved by fusing the characteristics of wavebands and automatic threshold algorithm. Further work will focus on the utility testing of the proposed model in some other satellite images, for example, Sentinel-2, ZY-2, and high spatial resolution data such as Worldview 4.

Comparison with machine learning algorithms, the overall accuracy of the ODRVI model was slightly inferior, but it had many advantages in large-scale, multi-temporal vegetation mapping and FVC estimation. These advantages can be summarized as follows:

(1) The computational complexity was low and the production efficiency was high. The spectral band's computation load of the ODRVI model is far less than the machine learning algorithms in both time and space complexity [28]. (2) The ODRVI model does not need sample selection and training. However, for machine learning algorithms, the quality and quantity of samples directly affected the accuracy of the algorithm [27]. (3) The ODRVI model was more conducive to the FVC estimation using the mixed pixel decomposition estimation model. The linear regression fitting degree was high between the ODRVI model and FVC estimation models.

### 5.2. Influence Factors of the ODRVI Model

Like other typical VIs, ODRVI achieves a division of ground object gray grade by bands combination calculation, which can improve the display of vegetation and suppressed non-vegetation display in index images. However, the accuracy of ODRVI is affected by the external environment, such as the quality of remote sensing images, spatiotemporal change, seasonal change, weather, and climate [53]. The results of vegetation extraction have differences derived from multi-temporal remote sensing images. In this experiment, the selected images were acquired during the lush vegetation period. During the vegetation growing season, the natural bare soil was covered by vegetation to reduce the interference for vegetation extraction. Moreover, surface temperature, rainfall, and sunlight have a crucial influence on vegetation growth. Suitable temperature, plentiful rainfall, and adequate illumination bring dense and healthy vegetation and improve FVC with areas increasing high and medium-high FVC levels. Conversely, the area of ELF and LF increased. For example, compared with 2000, in 2001, the area of total FVC decreased by 516.85 $km^2$, among them, the area proportion of MHF and HF FVC decreased by 5.08% and 4.05%, respectively. In 2017, compared with 2016, the area of total FVC decreased by 3888.42$km^2$, the area proportion of MF and MHF decreased by 27.56% and 7.51%.

Additionally, the performance of ODRVI is also related to the geographic spatial variation of images and the value of parameter $\theta$. Spatial variation leads to a difference in reflectivity of the same ground objects. The values of parameter $\theta$ change the index value of ground objects with variable discrimination, bringing a difference of accuracy. In this study, the same value of parameter $\theta$ was selected for three regions to verify the applicable performance of ODRVI in different spatial regions, which would lead to accuracy deviations of FVC in special regions. However, an automatic threshold adjustment algorithm can address the difference by selecting the optimal threshold for multi-temporal images [48].

### 5.3. Performance of FVC Estimation

The performance of FVC estimation is affected by the model sensitivity to the soil background, atmospheric, spatial scale, and sensor type [4]. The soil noise affects VI values due to soil properties and leads to FVC estimation deviation from the ground-truth value [54]. FVC in high density vegetation cover areas tends to be overestimated, while FVC in low density vegetation cover areas tends to be underestimated. In this paper, improved FVC estimation fused the ODRVI and the VCVP model. The results of the ODRVI were taken as input parameters and determined the accuracy of the FVC estimation. All of the FVC estimation factors mentioned above were reflected in the ODRVI model. The ODRVI model minimizes the influence from the soil background by parameter adjustment. Therefore, the improved FVC model reduced the influence from the soil background. Moreover, the proposed ODRVI model also reduced the influence of the atmosphere by a dynamic adjustment of the reflectance value of the NIR band. For the influence of spatial scale and sensor type, the ODRVI proposed an automatic threshold adjusting the algorithm to overcome the accuracy difference for multi-spatiotemporal images [48], and improving the stability of FVC estimation.

In experiments, we designed a comparison between the proposed FVC estimation and ground-truth value. The proposed FVC estimation was also compared with the DP-based FVC estimate in high- and low-density vegetation cover (see Figure 9). The

performance of the VCVP-based FVC estimation reduced the deviation from the ground true value by 4.38% and 11.27% than the DP-based FVC estimation in high- and low-density vegetation cover areas, respectively. Therefore, the VCVP-based FVC estimation decreased the overestimation of high density and the underestimation of low density in DP-based FVC estimation [35].

However, vegetation constitutes the fundamental part of the earth's ecosystem and provides diversified features with spatiotemporal change. The variable features bring a few challenges to estimating FVC from remote sensing images. Remote sensing images are derived from electromagnetic waves of ground object reflectance, and FVC presents vegetation structure, constructing a relationship between the two is crucial for FVC estimation [55]. The current FVC estimation models are limited by seasonal variation, solar illumination, climate change, and territory from remote sensing images.

*5.4. Influence of Urban Sprawl on FVC Change*

The total FVC area decreased by 3265.52 km$^2$ over the past 20 years (Figure 10). The urban area of Hefei has been continuously expanding, and the urban impervious surface area has increased by 504.88 km$^2$ compiled from historical economic development data (1999–2018). Urban sprawl reduced the vegetation cover area and changed the spatial distribution of each FVC level [56]. The impact of urban sprawl on FVC changes in Hefei can be summarized as follows:

(1) Urban development reduced the total area of FVC and increased the fragmentation of vegetation cover areas. The urban area expanded annually, and the fragmentation degree of vegetated areas started to increase as areas surrounding the downtown region were developed, such as in 2010, 2013, and 2014. In addition to seasonal change and weather influences, the increased fragmentation of vegetated areas occurred due to the expansion of the city and human activities.

(2) Urban sprawl changed the spatial distribution of all FVC levels. Urban expansion decreased vegetation cover and changed the area of FVC to different degrees. For example, the area of ELF, LF, and MF decreased by 13.39%, 6.35%, and 1.61%, respectively, in the southwest and southeast areas of Hefei from 2009 to 2013.

(3) Urban sprawl changed the area of all grades of FVC. Urban greening and afforestation slowed down the rate of FVC change to a certain extent. The area increased in the ELF and LF levels. The government deepened its understanding of environmental changes caused by urban development and ensured the protection and conservation of original forests and green spaces. The proportion of MF and HF increased. The proportion of HF increased by 0.18% from 1999 to 2018, and that of MHF increased by 3.4%.

(4) Urban sprawl accelerated urban water pollution and reduced the vegetation cover in the surrounding areas. An increase in the proportion of urban impervious surfaces caused a large amount of surface runoff. Sediment, rich nutrients, pesticides, and garbage entered the water, increasing water pollution and killing vegetation.

Urban FVC dynamic mapping is a crucial contribution to urban ecological environment quality, which substantially impacts the production and living standards of urban dwellers. Thus, the resulting FVC maps from the current analysis can act as a basis for urban planning policy formulation, and urban biodiversity conservation action development. Particularly, if we separate the urban forest change from the FVC change, we may assess the carbon source and sink characteristics, which further contributes to the studies on urban climate change and urban heat islands. In this paper, the ODRVI model and the VCVP-based FVC estimation can meet the need for urban FVC dynamic mapping from multi-temporal Landsat images.

**6. Conclusions**

In this paper, we developed an improved VI model and enhanced the accuracy of FVC estimation. The proposed ODRVI model and improved FVC estimation were used

to map the annual dynamics of FVC in Hefei from multi-temporal Landsat images. Our contributions were as follows:

(1) The ODRVI model was proposed to improve the sensitivity to the water content, roughness degree, and soil type by minimizing the influence of bare soil in areas of sparse vegetation cover. It improved the overall accuracy of vegetation extraction and the ability to distinguish vegetation from non-vegetation.

(2) The ODRVI enhanced the stability of FVC estimation in the near-infrared (NIR) band in areas of dense and sparse vegetation cover. The ODRVI model was verified to have better performance in multi-temporal Landsat images, and obtain higher accuracy than the typical VI models by dynamic threshold adjusting.

(3) An improved FVC estimation method based on the ODRVI model and VCVP-based model was proposed. The VCVP-based FVC estimation had a more stable performance than the DP-based FVC estimation in both high and low density vegetation cover areas, and the classification error was relatively low.

(4) Annual dynamic change mapping of FVC using the VCVP-based FVC estimation model was applied in Hefei over 20 years. The total FVC area had an overall decreasing trend, and the fluctuation change of all FVC grades was observed. The process of the fluctuation FVC change was divided into four stages: Continuous rapid decay (1999–2004), rapid decline with fluctuations (2005–2008), fluctuation attenuation (2009–2013), and fluctuated increase (2014–2018). Urban sprawl played a crucial role in the change of all FVC grades.

**Author Contributions:** Y.W. proposed and designed the ODRVI model and wrote the manuscript, conceived and performed the experiments and analyzed the experimental results. M.L. designed and revised the manuscript. All authors have read and agreed to the published version of the manuscript.

**Funding:** This work was financially supported by the National Natural Science Foundation of China under grant Nos. 31971577 and 41971415, Anhui Province University Natural Science Foundation and Chuzhou University Talent Foundation Project (Nos.: KJ2020A0717 and 2020qd31), and the PAPD (Priority Academic Program Development) of Jiangsu provincial universities.

**Institutional Review Board Statement:** Not applicable.

**Informed Consent Statement:** Not applicable.

**Data Availability Statement:** Not applicable.

**Acknowledgments:** The authors would like to thank the U.S. Geological Survey (USGS) for providing the Landsat surface reflectance datasets, and thank the Anhui Provincial Bureau of Statistics for providing the historical statistics datasets.

**Conflicts of Interest:** The authors declare no conflict of interest.

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
