# Peer review of "Annually Urban Fractional Vegetation Cover Dynamic Mapping in Hefei, China (1999–2018)"

_remotesensing, doi:10.3390/rs13112126_

Round 1

Reviewer 1 Report

I'm satisfied with the authors corrections, but I still have one major comment on the discussion section:

The authors presented the main findings without comparing them with the previous findings in other studies. In other words, the authors need to justify the findings of this study by comparing them with the results of earlier studies by other researchers.

Author Response

Response to Associate Editor’s and Reviewers’ Comments for Manuscript

(remote sensing- 1206493)

Thank you very much for having our paper entitled “Annually Urban Fractional Vegetation Cover Dynamic Mapping in Hefei, China (1999-2018)" reviewed and sending us a bunch of comments, which are quite helpful for us to improve the manuscript. We have carefully revised the manuscript in accordance with the comments raised in the peer-review process, and the original comments are presented in black and our corresponding point-by-point replies are presented in blue.

We have checked the entire sections of the manuscript including main text, figures, tables, and references to ensure its compliance with the style or format of Remote Sensing. All modifications according to reviewers’ comments are marked in blue ink in the revised version. The itemized response to each comment is provided as follows.

Response to Reviewer 1’s Comments

Overall Comments: I'm satisfied with the authors corrections, but I still have one major comment on the discussion section: 

Response: Thank you for your comments. We have improved the discussion section in accordance with your comments.

Point 1: The authors presented the main findings without comparing them with the previous findings in other studies. In other words, the authors need to justify the findings of this study by comparing them with the results of earlier studies by other researchers. 

Response 1:

We have added a performance comparison between the typical vegetation index models and the ODRVI model we proposed and also compared the correlations between the extracted results from these indices model and the FVC ground truths in the Results Section (See Section 4.2, page 12). Moreover, we substantially reorganized the Discussion section, including the model performance discussion, comparison with previous studies, advantages and disadvantages of the proposed model, and model application fields and further prospects. (See pages 17-20).

In closing, we would like to again thank you for your valuable suggestions and comments which helped us to significantly improve the technical contents and presentation of our paper. We look forward to your positive feedbacks.

Sincerely,

Yuliang Wang

Mingshi Li

Reviewer 2 Report

Annually Urban Fractional Vegetation Cover Dynamic Mapping in Hefei, China (1999-2018)

Line 17. For the Abstract section, do not explain to much, try to make it clear with short and concise sentences. At this Line, you are saying why did you propose the ODRVI model in a complex way, and you are not talking about the VCVP model.

Line 19. Which model? The ODRVI? VCVP? Or the one that you are fusing?

Line 44. “Land cover change” is better known like “Land Use/Land Cover” change, and this last integrates such land use cover as well as vegetation cover measurements. You can put in this line “Land Use Land Cover and applications” and immediately put the 13, 14 and 17 cites, deleting the words “vegetation cover changes” in Line 45. 

Line 46. Try this: “Moreover, the spatial distribution of FVC and its spatiotemporal changes…"

Line 71. Change the coma after “(RT)” and put a connective word (and, or, such).

Line 80. Remove “established”.

Line 88. Remove the “coma” (,) after the cite 34.

Line 91. Try this: “… may cause an underestimated FVC…”.

Line 92. It is not necessary to put the lines “- “in the sentence.

Line 95. using an index “adequately considering”? Improve the sentence.

Line 97. It is plenty known what is a multi-spectral image. So, it could be unnecessary the words between the parenthesis.

Line 99. Do not put too many indices, it is clear with only 4 or 5 mostly known.

Line 106. You are saying “advantages and disadvantages”, but you are only explaining like “disadvantages”.

Line 118. Put “dynamic changes of FVC”.

Line 119. During the introduction, it is not clear if you are creating and employing or just employing the ODRVI and VCVP models for this study. 

Line 127. “of” instead of “for”.

Line 140. The “west mountains”.

Line 147. About the sentence of “the spatiotemporal changes…” it is not relevant in this part, because it is only a description of the study area.

Line 149. The delimitation area with that red delimitation is hard to see. Try another color.

Line 155. Remove “of”.

Line 162. Remove “with differences in numbering of bands and wavelengths”. It is known that every sensor has different spectral resolution.

Line 176. Add “the” between “improve” and “interpretation”.

Line 177. Put a “coma” (,) after “data”, because is a too long sentence.

Line 189. If you are going to talk about “results”, do it, but in the “results” section, not here.

Line 192. You are detailing results in the methodology when you talk about the “overall performance. Try to change that.

Line 228. You are able to put the acronym “VPVC” instead “vertical porosity vegetation canopy”.

Line 230. Serve it similar to Line 228.

Line 234. It is confusing to put “In this paper, the OVDRI was calculated…” until this line. It could be in the first lines of this part (3.2) of the methodology.

Line 248. “resulting in difficulties using” is not understandable.

Line 260. You can put “… shows light or bright gray”.

Line 266. “by compared” or “comparing”?

Line 274. To indicate de figures, you can try this: “Figure 2 (a and b)…”.

Line 282. Attend it in a similar way like Line 274, about how to indicate Figures.

Line 285. Attend it in a similar way like Line 274, about how to indicate Figures.

Line 295. Change the color of those red words in the images that are pointing out a place, it is hard to see in some cases.

Line 296. It is complex to see the figure in multiple parts (for example, a part in a page, and the rest of the Figure in the next page). Try to put the whole Figure in just one page, it will be more comprehensible to the reader. Attend this for all the Figures where you are sectioning in different pages of this manuscript. 

Line 301. Figure 4 after its respective mention in the text, not before. It appears suddenly.

Line 318. When you talk about the CNN algorithm, you are explaining like a methodology. This section is for results. Consider this by structuring it as a result, or add it in to the methodology section.

Line 331. Attend it in a similar way like Line 274, about how to indicate Figures.

Line 338. Attend it in a similar way like Line 274, about how to indicate Figures.

Line 358. What does that “(2)” at the beginning of this line mean?

Line 380. What does the acronyms in the Table mean?

Line 391. Figure 7 after its respective mention in the text, not before.

Line 402. Attend it in a similar way like Line 274, about how to indicate Figures.

Line 429. Figure 9 after its respective mention in the text, not before.

Line 435. Try this at the beginning of this Line: “… all FVC levels”, like in Line 433.

Line 438. Put a “coma” (,) to separate 10000 km² to 10,000 km².

Line 438. Try this: “… indicating a “grow” of…”, because you are already using “increased”, and it seems to be repetitive.

Line 440. “FVC grades”, not “grades FVC”.

Line 501. “FVC levels”.

Line 541. This part seems more like a result than a discussion. You may be able to add a few cites to discuss this point.

Line 543. “notorious” instead of “significant”.

Line 557. Normally, the conclusion section serves to highlight the most notable results and findings, without the need to indicate data related to amounts, percentages, areas and other, this because it has already been shown at the results section. Therefore, improve this section not only by detailing the results, but detail the main contributions of this work.

Author Response

Response to Associate Editor’s and Reviewers’ Comments for Manuscript

(remote sensing- 1206493)

Thank you very much for having our paper entitled “Annually Urban Fractional Vegetation Cover Dynamic Mapping in Hefei, China (1999-2018)" reviewed and sending us a bunch of comments, which are quite helpful for us to improve the manuscript. We have carefully revised the manuscript in accordance with the comments raised in the peer-review process, and the original comments are presented in black and our corresponding point-by-point replies are presented in blue.

We have checked the entire sections of the manuscript including main text, figures, tables, and references to ensure its compliance with the style or format of Remote Sensing. All modifications according to reviewers’ comments are marked in blue ink in the revised version. The itemized response to each comment is provided in attachment.

Reviewer 3 Report

Compared to the last submitted version, the manuscript was improved. Its contents now describe details about the new method to map fractional vegetation cover. However, it still needs to be revised before publication, especially the discussion section:

  • Authors still put results in Discussion texts. Section 5.1 and 5.4 are mainly results rather than discussion. I recommend authors to put these outcomes into Results section and add more discussion content to the right section. For example,
    • In my humble opinion, Section 5.1 should explain why new VI is better for land classification in China’s cities, and could it be useful to apply on other areas/ regions? It would connect well with their 5.2 section.
    • Same things should be done in section 5.4 as well. This is discussion, not showing results.
  • Authors extracted quantified results from their maps, where they should be provided by tables. For example, audiences cannot extract 6.63% and 12.73% from figure 11 (5.4). Same problem happened with figure 5, and figure 8. Should authors provide a table with their quantified results to support their facts?
  • Authors added some references to their discussion texts, but there are not enough links with previous studies. They need to focus more on linking with similar studies to show/ prove what knowledge we could gain from their key results/ analyses.
  • There are some minor presentation issues as followed.
    • Date of the scene (figure 1), could it be 25 July 2016 (texts on the figure)? If yes, audiences might still be easy to miss that information.
    • Figure 2 and its caption has been separated. Same problem happened with figure 4.
    • A figure should be placed right after the text first mentioning it. However, there are figures (figure 4 and 9) placed before the first mentioned texts. Please be consistent while putting figures before/ after texts with their references.

Author Response

(The authors gave the same response as above.)

Round 2

Reviewer 1 Report

I am satisfied with the revised manuscript and recommend proceeding with publication.

Reviewer 3 Report

In general, authors have addressed all issues I mentioned. While the paper looks good, it might still need minor spell check to detect any potential spell or grammar mistakes. 

This manuscript is a resubmission of an earlier submission. The following is a list of the peer review reports and author responses from that submission.

Round 1

Reviewer 1 Report

The manuscript by Wang and Li examines the potential of an improved model for estimating fractional vegetation cover FVC called (optimized dynamic range vegetation index (ODRVI)). In general, the manuscript takes a worthy topic. However, before publication, there are some questions/revisions that need to be addressed.

Major comment:

The quality of English is quite poor in some places, which has led to some parts of the paper being difficult to understand. It is quite possible some of my major comments can be addressed by better explaining methods and re-phrasing arguments.

Introduction:

Major comments: The author’s needs to add information about how different spatial and spectral resolution may affect the retrieval of FVC!

Minor comments:

L104: Why NDVI is not suitable for FVC over 60%? I think it’s because of saturating issue! However, using different indices may fix this issue!

L105:  MSAVI is sensitive to low vegetation canopy as it developed to minimize the effect of soil background! Please correct this statement.

L102-111: How about the raw spectral data? For example, using the spectral bands as an independent variable to estimate or measure FVC? This information has to be added to the introduction section.

Methodology:

L 172: 173:  change this sentence to: The spectral vegetation indices were introduced to improve interpretation of vegetation signals when using remote‐sensing data and can be used to measure vegetation status and growth while minimizing solar irradiance and soil background effects.

L205 -211: Please add a relevant reference to this paragraph.

L275: I recommend changing the section title to results!

L 276: Performance of ODRVI: The comparison results between Landsat RGB and ODRVI do not tell anything special regarding FVC!  When we talk about performance, we need to see the difference in value, as presented in section 4.2. I recommend using a different approach for such comparison, for example, creating different zones and showing the result in a smaller box with descriptive statistic values.

Table 1: The authors present the most important accuracy information -; however, they could also show omission and commission error rates by using the sample points identified in L321- 323.

Figure 5:  My main concern in this figure is related to the gray map for ODRVI ! I think there will be no major difference when it is presented in color like what they did for CNNs.  

Discussion

The discussion section is too long, and its somehow repeating the result section. In this section, it is essential to focus on the main findings and compare it with the previous studies to identify the study findings' differences and importance.

Having said all this I do think there is the core of a well-founded empirical study here, but the editor’s decision will need to reflect the amount of time that will be required to make the manuscript suitable for publication.

Reviewer 2 Report

The authors this paper, developed a new VI to improve the accuracy of Fractional vegetation cover (FVC) estimation. The study is well designed and executed, described in detail, and the conclusions are sound.  This information is of high interest to researchers and managers working in urban sprawl.

My recommendation would be acceptance of the article for publication following these minor corrections.

Line 147: The map needs to be improved

Line 154: Figure 2. Figure 2 improve writing

Line 163. I suggest provide more information of geometrically and atmospherically corrected

Line 321: CNN or CNNs. Standardize in the manuscript

Reviewer 3 Report

I think the topic is appropriate and interesting for the Remote Sensing journal. The title and abstract are concise and clearly describe the contents of the paper. The presentation is clear and the language is fluent. Methods are clearly described and can be applied by other scientists. I believe the study could have potential because advances in these topics are important to remote sensing science. However, in this present form, the study represents a very modest contribution to our understanding, but I think that, if authors carry out a better organizational focus on the main ideas throughout the text and rewrite the study, it would help the article a lot and it would be very close to being an important contribution. I think the study hides more work than the one is presented. I hope that the enclosed comments will assist to authors in revising it for a possible publication.

- I think the Introduction chapter does not give a clear background of the problem to be addressed. 

- Please, try to clarify the objectives of the study. I think authors should do an explicit definition of the main objectives of the work or the problem to be addressed. The objectives should give a clear vision of the originality of the manuscript.

- I have not got clear an important question: why the study is important, i.e., what is its contribution to science? The novelty, originality, and scientific contribution of this study in its respective field are not well presented. I think the authors should clarify why the study is important? I think the authors should make clearer these points. This should give a clear vision of the originality of the manuscript.

- In general, the study is very long and has a lot of figures. Some chapters, such as methodology and results, should be shortened and rewritten.

- Authors have called Discussion an extension of the Results chapter. I think the discussion should be a different chapter. There is no debate between the results found by authors and the available literature. I think it is necessary to write again the discussion chapter trying to follow a storyline, in which authors compare their results with those from other studies.

- English is not my mother tongue, but I think the article should completely be revised. In general, the text is difficult to read in many sections because sometimes is redundant, so it is difficult to follow such an exposition.

Reviewer 4 Report

The article introduced a new vegetation index, optimized dynamic range vegetation index (ODRVI) to improve the accuracy of fractional vegetation cover (FVC) and showed the FVC analyses in Hefei, China over the past 20 years. The results showed that using ODRVI with the vegetation canopy vertical porosity (VCVP) model had a better performance than typical VIs in Hefei, Beijing, and Guangzhou. Each section’s content until the result one is detailed and clearly explained necessary information/ results. However, it contains major issues that need to be addressed before publication:

  • Historical meteorological datasets should be mentioned in “Study area and Datasets” rather than in Acknowledgment section.
  • Authors should explain their reasons to name sections like “Experiments and Verification” and “Discussion and Results”. The content of “Experiments and Verification” looks much like “Results” section while the remaining section presented both results (Ex: showing one figure) and discussion. It creates confusion for audiences including me.
  • The discussion contents did not totally match with results, especially section 5.2 and 5.3. Authors should present historical changes of meteorological and anthropogenic parameters before starting any discussion in Result section. Here, they discussed them without any prior information or result. Also, they (section 5.2 and 5.3) look like results rather than discussion.
  • I feel the manuscript attempted to focus on two different objectives but failed to connect these two to tell a complete and coherent story. Section 4 and 5 of this paper reflected it. Authors messed up between results and discussion, analysed out-of-nowhere results, hardly connected results of this paper with previous ones and explained their contribution or new knowledge to the remote sensing community. I feel that authors should focus on the first objective (new VI/ models to assess FVC), especially discussion's content and this manuscript would be more focused and become a better paper.
  • Some other minor problems:
    • There is an issue with the manuscript’s layout. I can see too wide left margin.
    • L147 (Figure 1 caption): It needs information about date and used bands (RGB?) of the Landsat image.
    • Also, should the figure 1 be placed after the first paragraph mentioned it? (L137)